# Heat Warnings in Switzerland: Reassessing the Choice of the Current Heat Stress Index

**DOI:** 10.3390/ijerph16152684

**Published:** 2019-07-27

**Authors:** Annkatrin Burgstall, Ana Casanueva, Sven Kotlarski, Cornelia Schwierz

**Affiliations:** 1Federal Office of Meteorology and Climatology MeteoSwiss, 8058 Zurich Airport, Switzerland; 2Meteorology Group, Department of Applied Mathematics and Computer Sciences, University of Cantabria, 39005 Santander, Spain

**Keywords:** temperature extremes, heat warning systems, heat stress index, impact-based warnings

## Abstract

High temperatures lead to heat-related human stress and an increased mortality risk. To quantify heat discomfort and the relevant dangers, heat stress indices combine different meteorological variables such as temperature, relative humidity, radiation and wind speed. In this paper, a set of widely-used heat stress indices is analyzed and compared to the heat index currently used to issue official heat warnings in Switzerland, considering 28 Swiss weather stations for the years 1981–2017. We investigate how well warnings based on the heat index match warning days and warning periods that are calculated from alternative heat stress indices. The latter might allow for more flexibility in terms of specific warning demands and impact-based warnings. It is shown that the percentage of alternative warnings that match the official warnings varies among indices. Considering the heat index as reference, the simplified wet bulb globe temperature performs well and has some further advantages such as no lower bound and allowing for the calculation of climatological values. Yet, other indices (e.g., with higher dependencies on humidity) can have some added value, too. Thus, regardless of the performance in terms of matches, the optimal index to use strongly depends on the purpose of the warning.

## 1. Introduction

Scientific evidence from the analysis of long-term climate records and future climate projections suggests an increase in the frequency and duration of extreme temperature events during this century [1,2,3]. The IPCC Special Report on Extremes ranks heat waves among the most severe risks associated with climate change [4]. It is further widely acknowledged that higher temperatures lead to heat-related human stress and discomfort [5,6], which is likewise expected to increase under a warmer climate (e.g., [7]), also in Switzerland [8]. Robust and effective heat warnings are essential to increasing people’s awareness of the risk of extreme heat situations [9]. With multiple societal sectors being in need of reliable warnings, an effective warning system is supposed to be flexible in order to account for variable demographical (children and the elderly as especially vulnerable groups) and occupational demands (indoor and outdoor activities with different levels of physical labor, clothing etc.) [10]. Additionally, regional-specific climatic conditions and phenomena associated with different levels of acclimatization among individuals, need to be taken into consideration [10,11]. In cities, people are potentially more affected by high temperatures than those in the surrounding areas, especially during night-time, as urban fabrics absorb and store more energy than rural areas, i.e., the Urban Heat Island—UHI—effect (e.g., [12,13]). Diversifying heat warnings is thus of major relevance for being able to issue user-tailored, impact-based warnings, that can guarantee the highest effectiveness. Within some meteorological projects or services, these requirements have been identified and systematically implemented already. The European project HEAT-SHIELD, for instance, includes the development of a personalized heat warning system particularly for occupational settings [14]. The underlying heat stress index wet bulb globe temperature (*wbgt*), which is solely based on meteorological parameters, allows for ISO-conformal guidance (i.e., according to the International Organization for Standardization, ISO) to reduce health impacts on the work force and to prevent associated productivity losses for the considered industries [15,16]. The diversified warning system of the German Meteorological Service (Deutscher Wetterdienst) represents another example. It combines perceived temperature (“Gefühlte Temperatur”) with daily minimum temperature inside buildings [17] to account for the typically higher heat stress in urban settings (UHI effect). User-tailored warnings are issued for all German municipalities exceeding 100,000 inhabitants [18]. Conscious of the special vulnerability of elderly people, the warning system is additionally adapted for high-risk groups [18].

Not all heat stress indices are based on meteorological data exclusively (so-called direct thermal indices; [19,20]) such as the heat index (*hi*) used in the heat warning systems in Switzerland and the United States (see Section 2; [21,22]). Some of them are based on objective and subjective strain (so-called empirical thermal indices; [19,20]) such as the physiological strain index (*PSI*; [23]), and still others further include physiological parameters and are thus able to account for the interrelation between metabolic activities, clothing and environmental parameters (so-called rational thermal indices; [19,20,24]). The universal thermal climate index (*utci*), for instance, is based on the heat balance of the human body. More precisely, it uses a multi-node human heat balance model, to include the meteorologically-induced heat stress a human body experiences when trying to maintain a thermal equilibrium with the outdoor surrounding [25,26,27,28]. The human heat balance model is augmented with a sophisticated clothing model, defining the effective clothing insulation and vapor resistance values for each of the thermo-physiological model’s body segments over various climatic conditions [29]. Although rational thermal indices are often based on highly advanced physiological models of the body’s response, simpler, direct thermal indices have certain advantages: they are especially suitable for long-term studies such as historical analyses of the climate [30] or climate projections. Other advantages are their less intense computational costs and higher flexibility. For instance, purely meteorologically-based indices only have to be computed once for a given meteorological setting, and can in a second step be interpreted or modified in terms of the specific individual setting (clothing, activity, grade of acclimatization etc.), which allows for user-friendly applications.

Direct thermal indices are often the preferred variables used to issue heat warnings in European countries, including Switzerland [31]. We present a case study on the comparison of the current heat warning system in Switzerland to a (virtual) new system that makes use of different and potentially more appropriate heat stress indices, which allows for a more flexible use and diversification in terms of specific warning demands. This study can be considered as a first step towards the general desire for expanding the current warning system to a more impact-oriented and diverse system, which, by definition, cannot rely on one single threshold or index, but needs to consider a broader set of user needs. The paper is structured as follows: First, we introduce the heat warning system in Switzerland. Second, the considered heat stress indices and data are described and we explain the employed methods. Third, we compare heat warnings based on different indices with the official heat warnings and analyze their potential matches. The paper ends with the conclusions.

## 2. Heat Warning System in Switzerland

The current heat warning system in Switzerland is based on the *hi,* which is purely meteorological, i.e., a direct thermal index, and combines temperature and relative humidity. The *hi* equation, derived from the apparent temperature [21], is a refinement of a result obtained by multiple regression analyses [22]. If daily maxima *hi* exceeds the Swiss-wide threshold of 90°F for at least three consecutive days, a heat warning of danger level 3 is issued. If the threshold of daily maxima *hi* of 93°F is surpassed for at least five consecutive days a warning of danger level 4 is delivered [32]. These are the warning levels currently in use in Switzerland. In terms of the criteria used for issuing the different levels of risk, the Swiss heat warning system does not take into account any epidemiological analysis of the relationship between thermal stress indicator and health impact (mortality). Instead, it relates to the US heat warning system, also based on *hi*, [33], which was adopted after summer 2003. The National Oceanic and Atmospheric Administration (NOAA; [33]) issues heat warnings in four categories, scaled in terms of possible heat-related health issues: when *hi* is expected to exceed a threshold of 80–90 °F (category “caution”), when it surpasses 90–105 °F (category “extreme caution”), when *hi* reaches 105–110 °F for two consecutive days (category “danger”) and when it surpasses 130 °F (category “extreme danger”). Note that these thresholds used in the US depend on the local climate and vary among different regions. As high danger levels (category “danger” and “extreme danger”) used by NOAA do not apply to Switzerland, MeteoSwiss relates to the lower risk level of 90 °F only and implemented an additional risk level of 93 °F. Compared to higher thresholds used in the US system, these better reflect extreme heat conditions prevailing at regional level in Switzerland.

*hi* is generally considered to perform well and offers robust results in the context of weather forecasting [8]. However, the index is rather inflexible and limited in its usage for further applications. Its lower bound at the level of 80 °F, for instance, does not allow for the calculation of continuous climatologies. Moreover, no associated ISO standards exist. It is exactly here where our analyses try to add important value. In the present case study, we compare the current heat warning system to a (virtual) new system that makes use of different and potentially more appropriate heat stress indices, which allows for a more flexible use and diversification in terms of specific warning demands.

## 3. Materials and Methods

### 3.1. Indices and Data

Eight widely-used heat stress indices are tested against the currently used *heat index* (*hi*) to issue official heat warnings in Switzerland, namely: wet bulb temperature (*wbt*; [34]), wet bulb globe temperature in the shade (*wbgt.shade*; [35,36]), wet bulb globe temperature in the sun (*wgbt.sun*; [36,37]), simplified wet bulb globe temperature (*swbgt*; [25,38]), apparent temperature (*apparentTemp*; [21,38,39]; corresponding to the shade apparent temperature following Morabito et al. [40]), effective temperature (*effectiveTemp*; [24,41,42]), humidity index (*humidex;* [38,43]) and discomfort index (*discomInd*; [24,44]). Depending on the considered index, different meteorological input parameters are required (Table 1): 2 m temperature (temperature), solar shortwave downwelling radiation (radiation), 10 m wind speed (wind) and/or relative humidity (relative humidity). A fifth input variable, the dew point temperature (dew point), is derived from relative humidity and temperature data. All indices (except *hi*) are provided in degrees Celsius and allow for the calculation of continuous climatologies. The *hi*, in contrast, is set to zero by definition below a value of 80. The *wbt* and *wbgt* (in the shade and in the sun) have been thoroughly tested and approved already in related contexts. The first has been used for heat stress analyses in the recently published CH2018 Swiss Climate Scenarios [45]. The *wbt* represents the temperature registered by a sensor that is covered with a wetted cloth. At 100% relative humidity the *wbt* equals the (dry bulb) air temperature. The *wbt* can be calculated by means of thermodynamic equations [46] or by using the empirical formula used in [34], as done for the CH2018 scenarios [45]. The *wbgt*, in contrast, is a widely used index in indoor (*wbgt.shade*) and outdoor (*wbgt.sun*) occupational environments. The *wbgt.sun* is a combination of the natural wet bulb temperature (measured with a wetted thermometer exposed to the wind and heat radiation at the site, it represents the cooling of the body via sweat evaporation), the black globe temperature (measured inside a 150 mm diameter black globe, it represents the heat absorption from radiation), and the air temperature (measured with a “normal” thermometer shaded from direct heat radiation). The *wbgt.shade* constitutes an approximation of the former, assuming no strong source of radiation and a constant value for wind speed of 1 m/s [25]. Both indices provide values that can be modified to account for clothing and offer standardized reference values for acclimatized and unacclimatized people with recommended rest/work cycles at different metabolic rates (ISO 7243: 2017) [16,47]. Detailed information and implementation on the remaining indices employed is provided in the referenced literature. All the heat stress indices have been calculated with the R package *HeatStress* (https://github.com/anacv/HeatStress, doi:10.5281/zenodo.3264929.).

The observational data used for the index calculation are provided by a network of 28 automated stations in Switzerland (Figure 1), operated by the Swiss national weather service MeteoSwiss (SwissMetNet stations) and in accordance with World Meteorological Organization (WMO) standards. Due to these standards, MeteoSwiss does not operate weather stations in urban areas. All sites are consistent with the official heat warning-relevant elevation of <600 m in Switzerland [48]. We use long-term (1981–2017) observational data to compute the indices at hourly resolution in a retrospective approach. To compare with the official warnings based on daily maxima *hi*, we aggregate the data and focus on maximum daily heat stress values exclusively. By investigating the daily maximum heat stress, there are no major differences between urban and rural sites as there are for daily minimum temperatures, due to the UHI effect (e.g., [12,13,49]), which justifies the use of high-qualitative rural station data.

### 3.2. Threshold Calculation

For testing the current heat warning system against a virtual new system that makes use of a different heat stress index, we investigate how well warnings that are based on *hi* match warning days (days above the threshold) and warning periods (consecutive days above the threshold) that are calculated from the alternative heat stress indices under consideration. In order to identify these warnings with the alternative indices, index-specific thresholds need to be determined. We test two different approaches: frequency-based approach (site-specific thresholds, i.e., different for every station) and averaging-approach (Swiss-wide thresholds, i.e., identical for all stations), to compare to the official Swiss-wide threshold of *hi* exceeding 90 °F [32]. Both threshold approaches can be likewise applied to the danger level 4 warnings by using the *hi* threshold of 93 °F [32]. In this paper, however, we focus on the more common danger level 3 warnings exclusively.

#### 3.2.1. Site-Specific Thresholds

Individual thresholds for each index and for each station are derived by using a frequency-based approach, meaning that the total number of official warning days at each station stays the same regardless of the index used. In more detail, we first identify the site-specific number of official warning days by counting the values exceeding the *hi* threshold of 90 °F. In a next step, site-specific values of the alternative indices are each sorted in ascending order. The one value that is followed by the exact same number of exceeding values as the number of official warning days is considered as the threshold value for the respective station and index.

#### 3.2.2. Swiss-Wide Thresholds

Official heat warnings in Switzerland are based on a *hi* threshold valid for whole Switzerland [32]. To ensure comparability, Swiss-wide and site-independent thresholds for each index are derived by using a straightforward averaging approach. To reduce the influence of extremes and to gain more robust results, all station values of the respective index for the period between 1981–2017 are pooled together and compared to *hi* values exceeding 90 °F (Figure 2). The index values that correspond to the positions of 88 °F < *hi* < 92 °F are averaged and the obtained value is considered the Swiss-wide threshold for the employed index (Figure 2). In contrast to the site-specific thresholds, the assumption of having the same total number of official warning days regardless of the index used is no longer valid for the Swiss-wide threshold. Instead, the total number of warning days at a given station based on an alternative index typically differs from the total number of official warning days.

### 3.3. Matches of Warnings for Single and Consecutive Warning Days

#### 3.3.1. Matches of Single Warning Days

We use a straightforward matching approach to identify the percentage of corresponding single warning days based on *hi* and the potential heat index alternatives. Official warning days are checked against days that indicate an index value exceeding the threshold (see Section 3.2) in order to examine the number of temporal matches. The sum of matching days relative to the total number of official warning days in the considered period 1981–2017 gives the overall percentage of matches (Figure 3). In order to obtain comprehensive site- and index-specific information on the percentage of matching single warning days, this approach is applied to each index and each station separately. The single-day matches calculation is performed based on the site-specific thresholds for the alternative indices (Section 3.2.1) to ensure the same frequency of warning days as for the *hi*.

#### 3.3.2. Matches of Consecutive Warning Days

Official level 3 heat warnings in Switzerland are based on at least three consecutive days above the *hi* threshold of 90 °F [32]. In line with this regulation, we identify for each index and for each station warning periods of at least three days in a row. To better account for the different temporal durations of the official warning periods, we distinguish between three-day events, four-day events and five-(or more) day events. By comparing the warning periods based on *hi* and the potential heat index alternatives (both using site-specific and Swiss-wide thresholds), we can identify and quantify potential temporal overlaps, which we refer to as matches (Figure 4).

Concerning these warning matches, two kinds of information are relevant for us (Figure 4): (i) The “hit rate” provides information on the number of official warning periods captured by the alternative index relative to the total number of official warning periods within the *hi* respective category (three-day events, four-day events and five- (or more) day events), regardless of the correct duration of the warning event (100% for four-day events in the example, since one out of one event is captured). Besides the number of captured warnings, the “hit rate” identifies potentially “missed” official warning periods within the respective category; that is, no warnings are issued by the alternative index, despite an official warning period. (ii) Whenever there is a hit (overlap of two periods for at least one day), the percentage amount of overlapping warning days compared to the duration of the respective official warning period of a category is calculated (“quality of hit(s)”, 25% in the example below, as one out of four days are captured). On the basis of these categories (three-day events, four-day events and five- (or more) day events), we then average the percentages within one category for each index and each station (only values exceeding zero are considered) to receive the mean station- and index-specific “quality of hit(s)”.

## 4. Results

### 4.1. Threshold Calculation

#### 4.1.1. Site-Specific Thresholds

Figure 5 shows the site-specific threshold anomalies, i.e., the local differences to the spatial mean, for all alternative indices considered as well as the spatial averages of the site-specific thresholds and the corresponding standard deviations.

The values obtained as reference thresholds change among indices as a result of the different input variables and their different combinations into a single number. Note, however, the partly pronounced spatial differences of thresholds within one index. This becomes obvious when considering the respective spatial standard deviations of the thresholds of *wbt* (standard deviation 0.49) or *apparentTemp* (standard deviation 0.60). In contrast, the site-specific thresholds of *swbgt* (standard deviation 0.05) or *discomInd* (standard deviation 0.09) vary markedly less and are thus more homogenous in space (Figure 5 and Table 2). The large spatial standard deviations of the site-specific indices are somewhat surprising since all index thresholds are derived by a frequency-based approach: by definition the number of days above the defined threshold of any alternative index and the number of days above the *hi* threshold are the same. The threshold of 90°F for *hi* has been applied to all stations. Thus, the threshold value of the respective index at “warmer” stations with higher *hi* values (i.e., more days above the threshold 90°F) is supposed to be similar to the threshold value of “colder” stations with lower *hi* values (i.e., less days above the threshold 90 °F). Hence, significant differences in absolute threshold levels within one index are not to be expected in the first place. However, since these considerations do not match the actual results of the majority of indices, we have proceeded with further analyses to better understand spatial threshold differences within a given index.

Large differences are noticeable, for instance, when comparing the absolute thresholds of *wbt* in Chur (21.5 °C) and Lugano (23.5 °C). We assume differing two-dimensional distributions of the input parameters temperature and relative humidity to be responsible for the differing thresholds of the two sites. A kernel density analysis allows for investigating the respective distributions for each station for the considered time period 1981–2017. The representation of the two-dimensional densities shows the probability of having different combinations of temperature and relative humidity (Figure 6). In Chur (Figure 6a), the histograms of both input parameters show unimodal-like patterns, whereas in Lugano (Figure 6b) the distributions (humidity in particular) feature two distinct maxima, resulting in different two-dimensional distribution patterns with two maxima in Lugano and only one maximum in Chur. The distribution of humidity in Chur, skewed to the right, is associated with a reduced probability of having high values of both variables at the same time, whereas in Lugano, relative humidity is multimodal with more weight on values of higher relative humidity, shifting the combined density towards warmer and moister conditions. These site-specific differences in distributional patterns, in addition to the differing mathematical combination of temperature and humidity in *hi* and the alternative index, are the reason for the spatial variation of the indices. The resulting distribution of *wbt* values in Chur (Figure 6c) and Lugano (Figure 6d) resembles very well the different distributional patterns of its input variables. In particular, temperature seems to be the dominant factor in determining the *wbt* distribution, pushing *wbt* in Lugano to higher density values in the upper tail. Local differences in thresholds result from differing distributions of input variables, i.e., localized meteorological characteristics. A country-wide threshold for such indices might overlook local characteristics related to different health risks.

#### 4.1.2. Swiss-Wide Thresholds

Official heat warnings in Switzerland are based on a single *hi* threshold valid for the whole of Switzerland. Table 2 contains the corresponding Swiss-wide thresholds for the alternative indices (1) obtained as the spatial average of the site-specific thresholds (and the spatial standard deviation, as in Figure 5) and (2) derived by the straightforward averaging approach described in Section 3.2.2. Values range between 22.0 (*wbt*) and 36.6 °C (*humidex*). Comparing the two approaches, the Swiss-wide thresholds show very similar results (Table 2, first and third column).

Both the employed frequency-based approach to calculate site-specific thresholds and the averaging-approach to calculate Swiss-wide thresholds compare well to previous works. Results for the mean site-specific and the Swiss-wide thresholds of *wbt* (22.4 °C and 22.0 °C) are supported by the Swiss climate scenarios [45] that use a *wbt* threshold of 22.0 °C. This threshold has been derived as a high percentile of the present-day distribution of daily values at Swiss measurement sites. Also, *apparentTemp* thresholds (site-specific threshold of 32.6 °C and Swiss-wide threshold of 32.2 °C) resemble very well the threshold as calculated in [5], that is 32.0 °C (although their implementation does not consider the effect of wind). The value as used in [5] corresponds to the 98th percentile of the pooled maximum daily *apparentTemp* distribution across the eight considered meteorological stations in Switzerland.

### 4.2. Matches of Warnings for Single and Consecutive Warning Days

#### 4.2.1. Matches of Single Warning Days (for Site-Specific Thresholds)

For all alternative indices, Figure 7 shows the percentage of matches of single warning days based on site-specific thresholds (Section 4.1.1) compared to official warning days based on *hi* values exceeding 90 °F.

Interestingly, among the indices, the percentages of matches differ considerably, with the spatial mean values ranging between 53.8 (*wbt*) and 95.5% (*swbgt*). To better explain the varying matching percentages among the indices and to identify the indices’ sensitivities towards their input variables, we conducted a response surface analysis (RSA, Figure 8). RSA is a statistical approach that examines the relationship between explanatory input variables and one (or more) response variable(s). By altering one input variable at a time, while the others remain constant, it is possible to identify the influence of each variable individually on the response variable. For an exemplary analysis we chose the indices with the lowest (*wbt*) and the highest (*swbgt*) matches, both based on the same input variables temperature and relative humidity. The results reveal similar sensitivities (slopes) towards temperature (Figure 8a,c), but in terms of humidity, *wbt* shows a stronger dependency than *swbgt* does (larger slope in Figure 8b compared to Figure 8d). We assume these differences in sensitivity to be responsible for the diverging percentages of matches. With *swbgt* yielding the highest matches, we further expect this index to best resemble the sensitivity of *hi* towards its input parameters. It is, however, important to mention that lower matches (e.g., *wbt*) should not be classified as poor results: Different sensitivities to the respective input variables might offer additional benefits for effective heat warnings, depending on the purpose of the warning.

The different sensitivities of the heat stress indices towards their input variables becomes apparent when considering the temporal evolution of the indices. As Figure 9 indicates (example of Basel, summer 2003), *hi* (Figure 9a) and *swbgt* (Figure 9c) seem to combine their input variables relative humidity and temperature in a very similar way, resulting in the exact same days above the respective thresholds (*hi* threshold of 90 °F and site-specific thresholds for alternative indices). This becomes clearer, when focusing on the first half of August 2003 when *hi* and *swbgt* show very high values for almost the whole period. In the case of *wbt* (Figure 9b), however, the high index values in the beginning of the period decrease much earlier (at the end of the first week of August 2003) than the *hi* and *swbgt* values do, leading to fewer values exceeding the *wbt* threshold during this period and thus to lower percentages of matches. Note that, by definition, the total number of warning days in the whole considered period of 1981–2017 stays the same regardless of the index used (see Section 3.2.1), meaning in total 119 warning days in Basel. Yet, zooming in to a shorter period results in differing numbers of warning days for the respective indices. In summer 2003, there are 17 official warning days, 5 *wbt*-based warning days and 17 *swbgt*-based warning days. In the case of *wbt*, warning days are less concentrated in summer 2003 and more widely spread among the considered period 1981–2017 than the *swbgt*-based warning days, that have the same temporal distribution as *hi* in this specific summer.

#### 4.2.2. Matches of Consecutive Warning Days (for Site-Specific and Swiss-Wide Thresholds)

We choose the stations Basel (BAS), Chur (CHU), Geneva (GVE), Lugano (LUG) and Zurich Fluntern (SMA) (see Figure 1) for an exemplary analysis of matching consecutive warning days based on site-specific (Section 4.1.1) and Swiss-wide (Section 4.1.2) thresholds. Picking these sites is motivated by our effort to include representative stations of different climate regions in Switzerland. We focus again on *swbgt* and *wbt*.

Figure 10 presents matches of overlapping warning periods based on *wbt* (site-specific thresholds) compared to *hi.* The alternative index *wbt* is not able to capture all official warning events (low “hit rate”). Yet, when there is a hit, the “quality of hits” is larger than 70% in most of the cases.

As expected, warnings based on *swbgt* (site-specific thresholds) show higher “hit rates” than warnings based on *wbt* (Figure 11), since for *swbgt* the matches of single warning days already show clear similarities with the official warnings (Figure 7). In general, all warning periods based on *swbgt* show an overlap with the official warning periods (“hit rate” of 100%). Also, the quality of overlapping warnings is very high: The majority of warnings based on *swbgt* is consistent with the official warnings with mean “qualities of hits” of 100%. Still, there is an interesting exception for Chur (CHU). For the five- (or more) day warning events, even two *swbgt*-based warning periods overlap with the official warning period (“hit rate” of 200%). Nevertheless, the mean “quality of hit(s)” is low (35%). Information on the matches of overlapping warning periods based on the remaining indices compared to *hi* (Appendix A) is provided as Appendix A.

Figure 12 illustrates the matches of overlapping warning periods in accordance with index-specific, but Swiss-wide thresholds for *wbt* (Table 2, third column). Bear in mind that, under this approach, the frequency of warning days is not necessarily the same for the *hi* and the alternative indices. Depending on the station, these matches (“hit rate” and “quality of hit(s)”) are higher (lower) compared to using site-specific thresholds, as a larger (smaller) number of days exceeds the warning level, which increases (reduces) the probability of overlaps with official warning periods. By focusing on selected stations (Figure 1) the pronounced station dependency of using Swiss-wide instead of site-specific thresholds becomes clear.

As seen in Figure 10, considering the category of four-day warning events in Lugano (LUG), for instance, one out of three official warning periods is captured when using *wbt* with its site-specific threshold of 23.5 °C (“hit rate”: 33%; mean “quality of hit”: 75%). Issuing warnings based on the *wbt* Swiss-wide threshold of 22.0 °C, as expected, raises the number of warnings in Lugano (from 111 to 509 days), due to a lowered threshold: According to Figure 12, three out of three official warning periods are then captured with a mean “quality of hits” of 100%. In Chur (CHU), in contrast, the individual threshold (21.5 °C) is lower than the Swiss-wide threshold (22.0 °C), resulting in a decreased number of warnings (from 70 to 33 days) and a reduced probability of overlaps. Consistent with our assumptions, the mean “quality of hit(s)” in the category of four-day events drops from 100% to 25%, with still capturing the official warning period. Yet, one needs to keep in mind that the change in the total number of warning days does not allow for a fair comparison in terms of the influence of using a site-specific or a Swiss-wide threshold, as the case of Lugano confirms: the higher “hit rates” and “quality of hits” stem from the higher total number of warning days when using a Swiss-wide threshold compared to a site-specific threshold, as it increases the probability of overlaps. Moreover, a large number of total warning days might also derive into many warning episodes which are not registered by the *hi*, increasing the so-called false alarm rate in the field of verification of weather forecasts.

This difference in the number of warning days is especially pronounced for indices with highly inhomogeneous site-specific warning levels like *wbt* (mean site-specific threshold 22.4 °C, standard deviation 0.49, Swiss-wide threshold 22.0 °C; Table 2) and less pronounced for indices with rather homogeneous site-specific thresholds like *swbgt* (mean site-specific threshold 25.5 °C, standard deviation 0.05, Swiss-wide threshold 25.4 °C; Table 2). For the latter, the total number of single warning days shows only minor changes between site-specific (WD_site_) and Swiss-wide thresholds (WD_CH_): Basel (WD_site_ #119 vs. WD_CH_ #131), Chur (WD_site_ #70 vs. WD_CH_ #71), Geneva (WD_site_ #96 vs. WD_CH_ #101), Lugano (WD_site_ #111 vs. WD_CH_ #136) and Zurich (WD_site_ #31 vs. WD_CH_ #38). The analysis of *swbgt*-based warning periods using the Swiss-wide threshold thus presents the exact same results for the considered five stations as the analysis based on site-specific thresholds does (Figure 11).

Still, the considered performance metrics (“hit rate” and “quality of hits”) are most informative when interpreted in a setting with site-specific thresholds in which, by definition, the total number of warning days for each index stays the same compared to the official number of warning days.

## 5. Summary and Conclusions

The current heat warning system in Switzerland is based on the heat index, *hi*, which performs well and offers robust results [8]. However, the index is rather inflexible and limited in its use for further applications: it presents a lower bound and does not allow for the calculation of continuous climatologies. Furthermore, no associated ISO standards exist. The desire towards a more flexible and impact-oriented heat warning system is related to the use of several thresholds or even several indices to be able to meet the large diversity of user demands.

In this paper, we compare the current heat warning system to a (virtual) new system that makes use of a different and potentially more appropriate heat stress index. For our analysis we consider eight widely used heat stress indices, all of them provided in degrees Celsius and subject to continuous distributions with no systematic lower bound. The respective single warning days and warning periods derived by the computed thresholds of each index are then analyzed in terms of potential matches or overlaps with official warnings.

For single warning days using site-specific thresholds, our results indicate that the percentage of matches varies considerably among indices. The different sensitivities towards the indices’ input variables might be the reason for the differing matches among the analyzed indices. We assume the index that best resembles the dependencies of *hi* towards its input variables offers highest matches. Among the employed indices, *swbgt* shows the highest (95.5%) spatially averaged matches, followed by *discomInd* (90.3%) and *effectiveTemp* (86.2%). *wbt* offers the lowest spatially averaged matches (53.8%).

In terms of consecutive warning days, two types of information are relevant for matches of warning periods: the “hit rate” (the relation of the absolute number of official warning periods that can be captured—at least one day of overlap—by the potential heat index alternative and the total number of official warning periods) and the “quality of hit(s)” (the mean percentage amount of overlapping warnings within one period relative to the duration of the respective official warning period).

When using *wbt* (site-specific thresholds), not all the official warnings are captured (low “hit rate”). Yet, when there is a hit, the “quality of hits” is larger than 70% in most of the cases. *swbgt* (site-specific thresholds) offers very high “hit rates” and “quality of hit(s)” among all warning categories, as already indicated by high matches of single warning days.

The influence of using a Swiss-wide threshold rather than a site-specific threshold, however, cannot be estimated in terms of the proposed performance metrics, as the total number of warning days at a given site may change substantially, when changing between the two employed thresholds (e.g., for *wbt*).

To summarize, *swbgt* resembles official heat warnings very well, both for individual and consecutive warning days. It even has further advantages: With having no lower boundary, the index allows for the calculation of continuous climatologies. However, other indices with higher dependencies on humidity like *wbt*, could give some additional information as well, despite their lower matches to the official warnings. Buzan et al. [38] confirm the different dependencies of heat stress metrics on their input variables: some may have a stronger dependency on regionally extreme temperature, while others have a stronger dependency on regionally extreme humidity. Through its major role in affecting the body’s hydration state humidity has a large effect on the heat-related human stress level: once the air temperature is higher than the skin temperature, the body can no longer cool via direct heat exchange, but needs to start evaporating, which is most efficient when a pronounced humidity gradient is present between the skin surface and the surrounding air. Otherwise, the body core temperature rises, causing severe health effects like heat exhaustion, heat cramps or even strokes [50]. An index placing weight not only on temperature but also strongly on humidity, could therefore be an advantage especially in heat-related human stress research areas.

Thus, regardless of the performance in terms of matches, the optimal index to use strongly depends on the purpose of the warning. According to Coccolo et al. [24], users need to choose the best thermal stress index according to their research’s needs, as a perfect solution to quantify the outdoor human comfort does not exist. Aiming at an effective, integrated and impact-oriented warning system, the combination of several indices, or at least different user-tailored thresholds depending on the specific warning demand, needs to be considered.

In most European countries the respective heat-health warning system (HHWS) is based on several thresholds at two to four risk levels, using single- or multiple-parameter methods, i.e., they use a single metric of temperature or multiple variables combined in one index [31]. The cited work [31] also found out that, in most cases, the warning levels are determined from epidemiological studies, linking heat to mortality (or other health indicators). Those relationships are established through statistical models, such as the distributed lag nonlinear model [51]. As the Swiss heat warning system does not take into account any epidemiological analysis, a way to improve the warning system could be to use a thermal stress indicator or heat stress indicator and to perform epidemiological analyses to identify location-specific thresholds to be then implemented operationally, building upon previous works such as [5]. This is important because, as described in a recent report of WHO-WMO on HHWS [9], usually the thresholds are response-specific, that means, the threshold values are determined at a level related to negative human response as suggested by the long-term relationship between some heat stress indicator and mortality. The threshold for action could then be the index value at which mortality (or any other health outcome) starts to rise substantially. Although this work is devoted to the heat warning system in Switzerland, it could serve as an example of application to provide support in the revision process of any heat warning system and paves the way to further reassessments of heat warnings.

## Figures and Tables

**Figure 1 ijerph-16-02684-f001:**
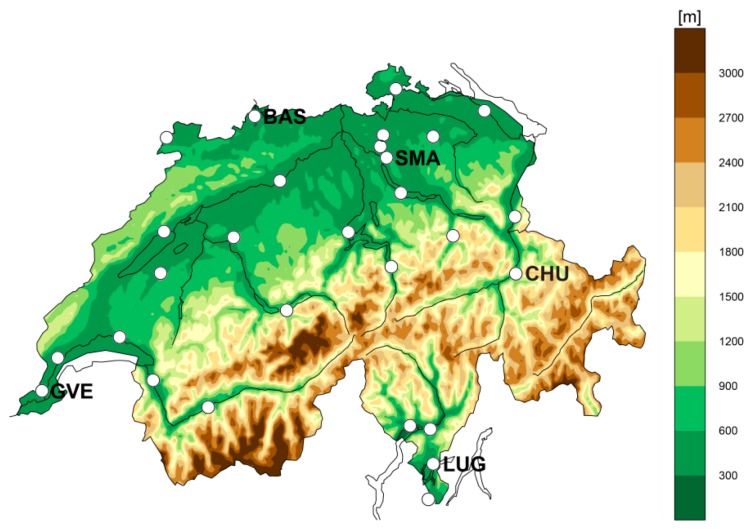
Location and elevation of the 28 considered automated stations of the Swiss national weather service MeteoSwiss (SwissMetNet stations). The selected sites are consistent with the heat warning-relevant elevation of <600 m [48]. The stations Basel (BAS), Chur (CHU), Geneva (GVE), Lugano (LUG) and Zurich Fluntern (SMA) serve as exemplary sites for later analyses.

**Figure 2 ijerph-16-02684-f002:**
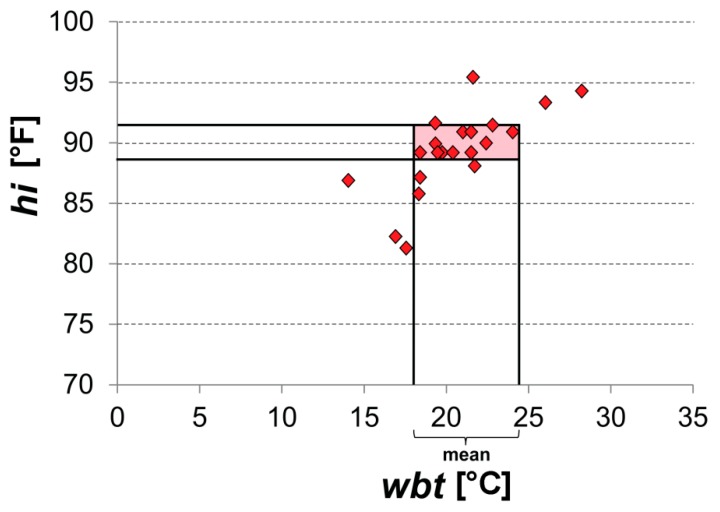
Exemplary Swiss-wide threshold calculation for wet bulb temperature (*wbt*) on the basis of 88 °F < *hi* < 92 °F. The markers represent virtual *wbt* and heat index (*hi*) pairs for all stations together.

**Figure 3 ijerph-16-02684-f003:**
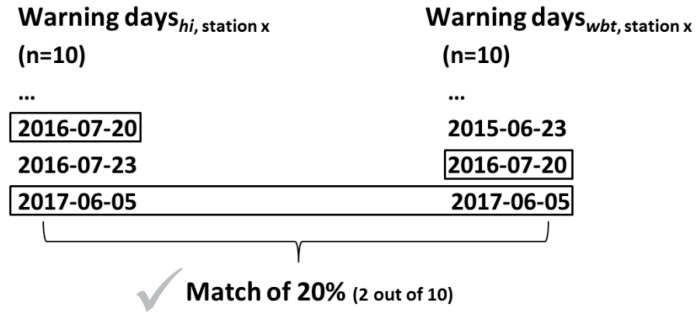
Exemplary matching of warning days (*n* = 10) based on *hi* (left) and *wbt* (right) for a station x. The black boxes mark matching warnings. Two out of ten *wbt*-based warning days match the official *hi*-based warning days, resulting in an overall 20% match using *wbt* in station x.

**Figure 4 ijerph-16-02684-f004:**
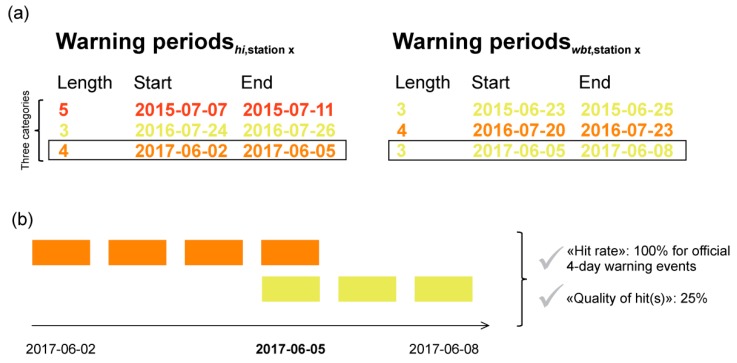
Exemplary warning periods for a station x, based on *hi* (left) and *wbt* (right). The warning periods are separated into three-, four- and five- (or more) day events according to their lengths. The black boxes mark warning periods that offer at least one day of overlapping warnings (**a**). In this example, the official four-day warning event is captured by one overlap (5th June 2017) when using *wbt*, which results in a “hit rate” of 100% in the 4-day event category and a “quality of hit(s)” of 25% (**b**).

**Figure 5 ijerph-16-02684-f005:**
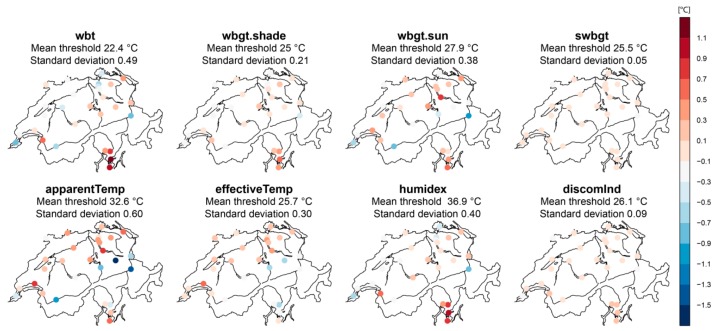
Spatial anomalies of site-specific thresholds (color-coded) and spatial averages and standard deviations of the site-specific thresholds (on top of each map) for the considered indices wet bulb temperature (*wbt*)*,* wet bulb globe temperature in the shade (*wbgt.shade*), wet bulb globe temperature in the sun (*wgbt.sun*), simplified wet bulb globe temperature (*swbgt*), apparent temperature (*apparentTemp*); effective temperature (*effectiveTemp*), humidity index (*humidex*) and discomfort index (*discomInd*). Anomalies are computed wrt. the respective spatially averaged site-specific threshold. Blue colors apply to negative threshold anomalies, red colors to positive threshold anomalies.

**Figure 6 ijerph-16-02684-f006:**
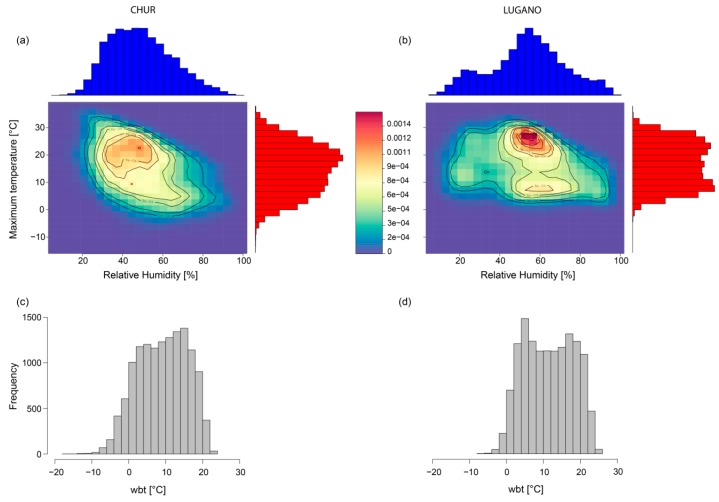
The 2-dimensional Kernel density plots of temperature and relative humidity (**a**,**b**) and the resulting distribution of *wbt* (**c**,**d**) for Chur and Lugano for the period 1981–2017. In Figure 6a,b, the top and right axes show the histograms of temperature (red) and relative humidity (blue). The density plots combine the distributions of both variables with red colors referring to higher probabilities and blue colors to unlikely conditions.

**Figure 7 ijerph-16-02684-f007:**
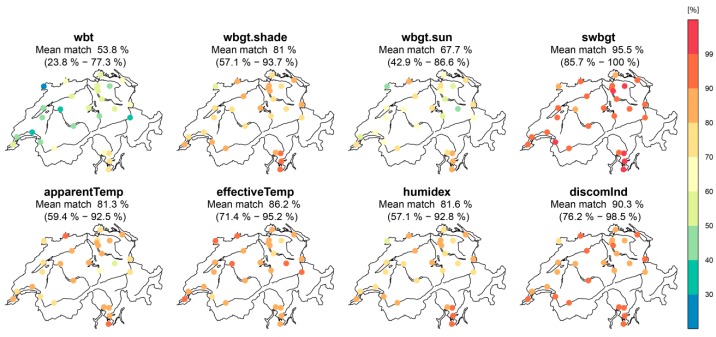
Percentage of matches of single warning days based on site-specific thresholds compared to official warning days based on the *hi* exceeding 90 °F, analyzed for the considered indices *wbt, wbgt.shade, wbgt.sun, swbgt, apparentTemp, effectiveTemp, humidex, discomInd*. Blue colors apply to lower matches, red colors to higher matches.

**Figure 8 ijerph-16-02684-f008:**
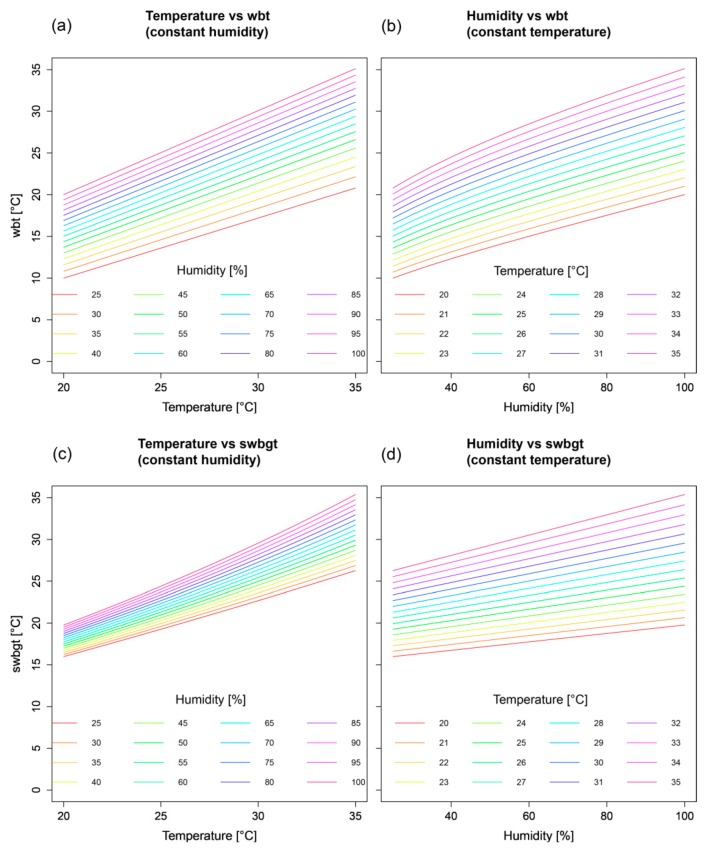
Response surface analysis of *wbt* (**a**,**b**) and *swbgt* (**c**,**d**).

**Figure 9 ijerph-16-02684-f009:**
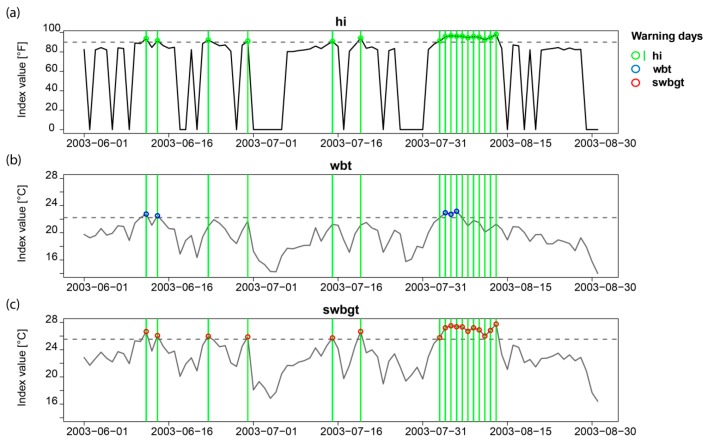
The daily evolution of *hi* (**a**), *wbt* (**b**) and *swbgt* (**c**) in Basel in summer 2003 and the respective warnings days: Green vertical lines (**a**–**c**) and green dots (**a**) show the official warning days based on *hi*, blue dots indicate warning days based on *wbt* (**b**) and red dots show warning days based on *swbgt* (**c**). Dashed horizontal lines mark the site-specific thresholds of each index.

**Figure 10 ijerph-16-02684-f010:**
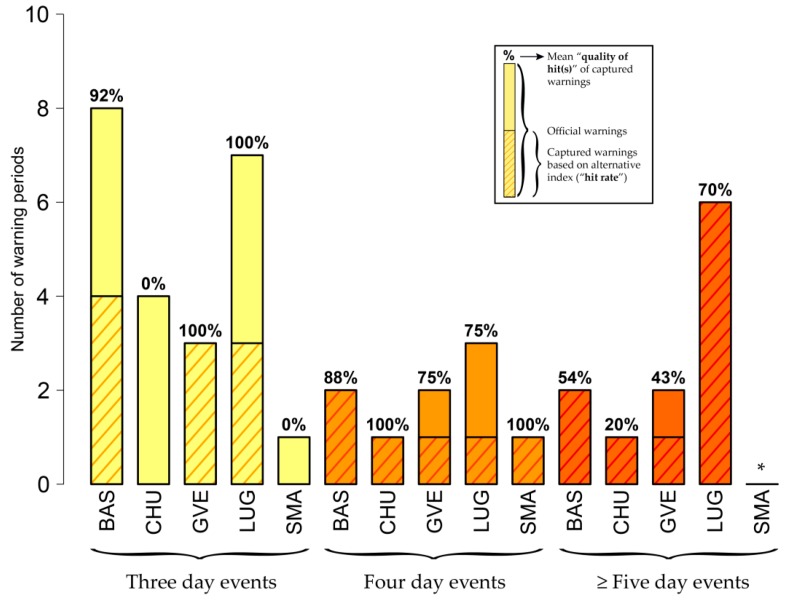
Matches of overlapping warning periods based on *wbt* (site-specific thresholds) compared to *hi.* The warning periods result from the use of site-specific thresholds (i.e., fixed total number of warning days for *hi* and every alternative index, WD_site_, see Section 3.2.1): BAS (WD_site_ #119), CHU (WD_site_ #70), GVE (WD_site_ #96), LUG (WD_site_ #111) and SMA (WD_site_ #31). The figure presents matches for three- (yellow), four- (orange) and five- (or more, dark orange) day official warning events. The shaded bars give the number of official warning periods of the respective duration. The dashed bars show the number of warning periods captured by *wbt* (indicating the “hit rate” when compared to the length of the entire bar). The percentage number on top of each bar gives the mean “quality of hit(s)” (Section 3.3.2). In Basel (BAS) and for the case of three-day warnings, for instance, four warning periods based on *wbt* show an overlap with (in total) eight official three-day warning events (“hit rate” of 50%). In the mean, these overlapping warnings show a 92% match (“quality of hits”). The asterisk applies to stations without any official warning periods in the respective category.

**Figure 11 ijerph-16-02684-f011:**
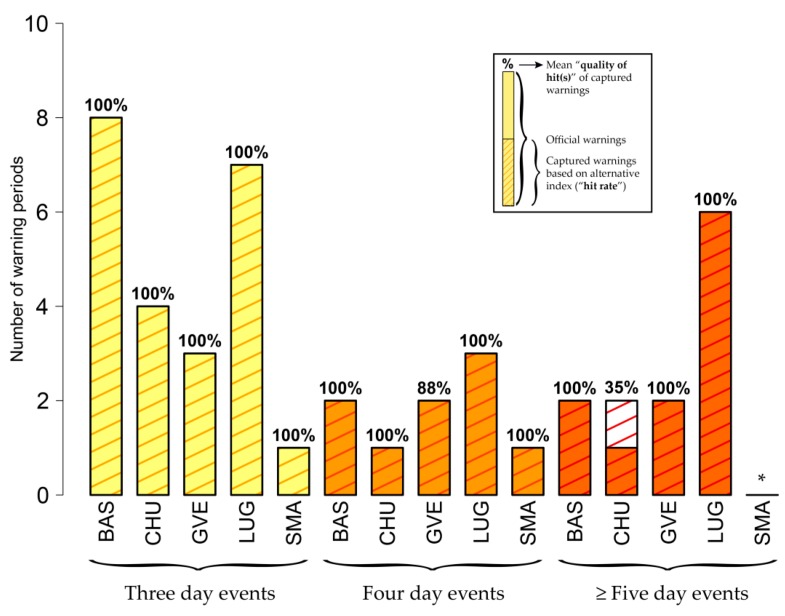
As Figure 10 but for *swbgt*.

**Figure 12 ijerph-16-02684-f012:**
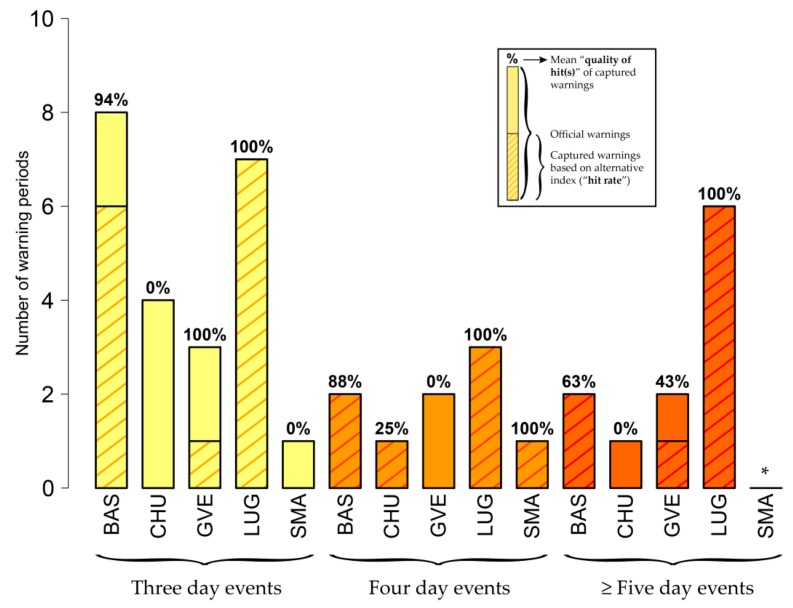
Matches of overlapping warning periods based on *wbt* (Swiss-wide threshold) compared to *hi.* The warning periods result from the use of Swiss-wide thresholds (i.e., the number of warning days is different for the *hi* and the alternative index, see Section 3.2.2). Compared to the number of warning days calculated with the site-specific threshold (WD_site_), the number of warning days changes individually for each station when using a Swiss-wide threshold (WD_CH_): BAS (WD_site_ #119 vs. WD_CH_ #154), CHU (WD_site_ #70 vs. WD_CH_ #33), GVE (WD_site_ #96 vs. WD_CH_ #53), LUG (WD_site_ #111 vs. WD_CH_ #509) and SMA (WD_site_ #31 vs. WD_CH_ #38). The figure presents matches for three- (yellow), four- (orange) and five- (or more, dark orange) day official warning events. The shaded bars give the number of official warning periods for the respective duration. The dashed bars show the number of warning periods captured by *wbt* (indicating the “hit rate” when compared to the length of the entire bar). The percentage on each bar gives the mean “quality of hit(s)”. In Basel (BAS) and for the case of three-day warnings, for instance, six warning periods based on *wbt* show an overlap with (in total) eight official three-day warning events (“hit rate” of 75%). In the mean, these overlapping warnings show a 94% match (“quality of hits”). The asterisk applies to stations without any official warning periods in the respective category.

**Table 1 ijerph-16-02684-t001:** Employed heat stress indices with respective input parameters.

Heat Stress Index	Temperature	Radiation	Wind	Relative Humidity	Dew Point
*hi*	x			x	
*wbt*	x			x	
*wbgt.shade*	x				x
*wbgt.sun*	x	x	x		x
*swbgt*	x			x	
*apparentTemp*	x		x	x	
*effectiveTemp*	x		x	x	
*humidex*	x			x	
*discomInd*	x			x	

**Table 2 ijerph-16-02684-t002:** Considered heat stress indices with respective mean site-specific (including the spatial standard deviation) and Swiss-wide threshold.

Heat Stress Index	Site-Specific Threshold (°C)		Swiss-Wide Threshold (°C)
	Mean	Standard Deviation	
*hi*	^-^	^-^	32.2 *
*wbt*	22.4	0.49	22.0
*wbgt.shade*	25.0	0.21	24.8
*wbgt.sun*	27.9	0.38	27.6
*swbgt*	25.5	0.05	25.4
*apparentTemp*	32.6	0.60	32.2
*effectiveTemp*	25.7	0.30	25.4
*humidex*	36.9	0.40	36.6
*discomInd*	26.1	0.09	26.0

* or 90 °F.

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
