# Peer review of "Heat Warnings in Switzerland: Reassessing the Choice of the Current Heat Stress Index"

_ijerph, 2019, doi:10.3390/ijerph16152684_

Reviewer 1 Report

The paper I really interesting, working on the thermal comfort at the Swiss scale.

I propose just some small improvements:

We are living in a urban world, and the stations are mostly located outside the city (as far as I know) if it is the case, please include a comment on the differences between the site of measurement, and where people live.

Please, include some comments on the future trends, as the heat stress periods are increasing in the next years, due to climate changes.

What can be the real application of this study? A mobile application for users? Please, comment it.

Reviewer 2 Report

This paper tests the comparability of various heat stress indices with the heat index (hi) used to issue official heat warnings in Switzerland.  The authors have found that the index simplified wet-bulb globe temperature (swbgt) has a similar sensitivity to hi in identifying days in which warnings are issued, except that the threshold level for issue of warnings is 25.40C compared with 32.20C for hi. The swbgt index is also similar in sensitivity and quality of hi in matching heat spells of 3-5 days duration.  The swbgt thus appears to be superior to the other indices tested in matching predictions with those of hi of days when the threshold for issuing warnings is exceeded.  Since hi is restricted by a lower bound of 800F, swbgt is suggested as a more flexible alternative. 

The similar performance of swbgt and hi is due to the fact that they have the same component input variables, shade temperature and humidity, and are stated to be combined in a similar fashion, but the article does not tell us how the inputs are combine for hi, but  for swbgt referral to the source article (ref #21) shows that swbgt is computed as:

sWBGT = 0.56TC + 0.393eRH 100 + 3.94

where eRH is vapor pressure in mb and TC is temperature in 0C.

The term swbgt appears to be inappropriate, since unlike wbgt it has no component of radiant temperature   (this is not the fault of the authors).  This suggests that, while the swbgt may perform in a similar way to hi, neither index is the optimal predictor of days when there will be heat related deaths or hospital admissions. The authors seem to acknowledge this when they state that indices other than swbgt and wbt may nevertheless be useful depending on the purpose of the warning.

There should be more information relating to the hi.  First, there should be some information on how the function is derived from its constituent parameters.  Secondly there should be more information on how effective it is as a predictor of increased death and mortality – the sensitivity and specificity. The authors state that hi performs well and offers robust results, but this statement is not referenced.  Elsewhere in the text it appears that references #4 and #23 are studies of the efficacy of hi.  Reference #23 is not accessible.  Reference #4 is a study showing a RR for mortality of 1.12 where maximum apparent temperature (Tappmax) was in the 98th percentile of all summer temperatures. The Tappmax happened  to be 320C, which is the same as the threshold level of hi used by the authors, but it is not clear whether the apparent temperature in the study reference #4 is the same as the apparent temperature as defined in the submitted paper, or whether it is the same as hi.  If the former, we have no evidence of the predictive value of hi other than the word of the authors that is delivers robust results.

It would also be helpful to know whether the hi as employed in this study is used by any other countries: otherwise the information in this paper is not of great practical value to readers outside Switzerland.

Although the conclusions drawn by the authors appear sound, this is a long paper with bewilderingly complex statistical computations, which may deter potential readers other than those with both an interest in the effects of climate on human health and endowed with knowledge of advanced statistical methods.

Table 1: Shade wbgt has a factor for radiation (via globe temperature)

Figure 2:  What is each red marker derived from? 20 data points but there are 29 monitoring stations?

Figure 5:  Here the term anomalies is introduced for the first and only time.  The meaning of anomaly in this context is not explained.  Presumably a negative threshold anomaly means that the threshold for that location is lower than the mean for that index.  If that is so, the example from Lugano in Figure 3 would lead to an expectation that Lugano would be marked in blue, since Figure 5 showa a sensitivity of only 20%: yet the colour of the dot is possibly black.

211-246: The authors have noted differences in thresholds of the same index for different sites, which for some indices is quite marked.  They conclude that this is due to differing distributions of input parameters (eg temperature and humidity) that comprise a single index value. This could be significant if the differing contributions to the index affect the health outcomes.  If this is so it would suggest that the index in question is not useful  in predicting outcomes: the risk to health (eg as measured by mortality rate) might better be evaluated by multiple regression than by using indices.

301-301:  The authors infer that findings for hi and swbgt are similar because they combine their input variables in a similar way.  They infer this from their analyses, and since the computation of swgbt is known (ref #21), this seems that the authors themselves did not know a priori the mathematical formulation of hi. The question must be asked: why is the derivation of hi not explained?

Reviewer 3 Report

Manuscript ID: ijerph-539787-peer-review-v1

Title: Heat warnings in Switzerland: Reassessing the choice 2 of the current heat stress index

I read this paper with great interest. The paper concerns a relevant topic that is the climate change and in particular the effects of heat waves and high temperature on human thermal stress. This cause an increased on mortality risk and this topic will become more relevant in the next years. The aim of the study is to identify the highest performing biometeorological index to be used in Heat Health warning systems in Switzerland. The research is very interesting, well conducted and the work is comprehensive. However, I still need some clarifications and maybe some changes could further improve it.

In particular, I believe that in the introduction a hat should be made where it is said that the Swiss heat warning system does not take into account any epidemiological analysis of the relationship between thermal stress indicator and health impact (mortality). This is important because, as described in a recent report of WHO-WMO on heat-health warning system (HHWS), usually the thresholds are response-specific, that is, the threshold values are set at a level associated with a negative human response as indicated by the long-term relationship between some measure of heat stress (for example, the Heat Index) and mortality. In HHWSs based on a “simple index” of heat stress, the threshold for action is usually the index value at which mortality (or any other health outcome) starts to rise rapidly, with the type of action or level of alert being determined by the intensity and duration of the period of exceptional heat.

Can you provide a section where the Swiss heat warning system is clearly described? In this way, also defining the criteria adopted for issuing the various levels of risk. I found something to the lines 132-134; lines 173-174; Lines 253-258. However, it might be better to bring them all together in the same section.

Why UTCI was not included in the analyses? UTCI represents the state-of-the-art in outdoor thermal comfort assessments. This would be very interesting since the aim is to test a “virtual” new system by using a more appropriate heat stress index. UTCI has the advantage of not having application limits (no lower bound) and above its output is expressed in °C. Please provide a description that justifies your choice.

Following several minor suggestions that could improve the manuscript:

Line 27: I suggest to cite https://doi.org/10.3390/atmos8070115

Line 32: I think it is: … demographical variables (children and      the elderly …

Line 40: … includes the development of a personalized heat      warning system …

Line 41: I suggest do not say that the WBGT is a type of heat      index, so as not to be confused with the index for the assessment of heat      stress "heat index". I would rather talk about the index for the      evaluation of heat stress (or thermal stress in general)

Line 42: Is the term “perceived temperature” mentioned here the      rational thermal index? 

Line 52: I think you referred to rational indices that are      indices based on calculations involving the heat balance equation.

Line 56: Please add that the UTCI also includes a sophisticated      clothing model that defines in detail the effective clothing insulation      and vapor resistance values for each of the thermo-physiological model’s      body segments over a wide range of climatic conditions.

Line 56: Please, Check out the references related to UTCI. I      suggest:

ü  Jendritzky G, de Dear R, Havenith G. UTCI--why another thermal index? Int J Biometeorol 2012;56:421-8.

ü  Havenith G, Fiala D, Błazejczyk K, Richards M, Bröde P, Holmér I, Rintamaki H, Benshabat Y, Jendritzky G. The UTCI-clothing model. Int J Biometeorol. 2012 May;56(3):461-70.

ü  Fiala D, Havenith G, Bröde P, Kampmann B, Jendritzky G. UTCI-Fiala multi-node model of human heat transfer and temperature regulation. Int J Biometeorol. 2012 May;56(3):429-41. doi: 10.1007/s00484-011-0424-7.

Line 64: please, add a reference for HI. Is it the “American” (NOAA)      Heat Index (https://www.wpc.ncep.noaa.gov/html/heatindex.shtml)      derived by the Apparent temperature index?

Lines 85-88: The bibliographical references do not refer to the      authors who developed the indexes, apart from Steadman for the apparent      temperature index. I suggest to include the exact (original) references      related to the indices used.

Table 1: Are you sure that WBGT.shade does not need wind?

Line 107: I think that both indices provide values that can be      modified to account for clothing …

Line 125: … the current heat warning system against …

Lines 125-134: It would also be interesting to include an      intensity indicator: of how much the threshold is exceeded. In this way      all the main characteristics to identify a heat wave will be considered:      the number of heat wave days (days above the threshold); persistent heat      wave (consecutive days above the threshold); the intensity of a heat wave      (how much the threshold is exceeded). These characteristics were included      in the manuscript I suggested to cite on line 27.

Lines 283-289. I think it is better to include the RSA      statistical approach in section 2.3.1. (materials and methods).

DISCUSSION SECTION: I suggest avoiding the methodology already      well described in the previous sections and focusing on the discussion of      the results and above all on the performances of the various indices.      Comparing the results obtained with other works could be very useful, thus      including some references.

If it is possible, try to include some comparison with other heat      warning system operational in other European countries which use similar      thermal stress indicators.

Line 453: after the colon, start with the lower case letter

It could be also interesting to include some part where you      describe what should be done to improve the Swiss heat warning system. For      example, use some other thermal stress indicator or heat stress indicator      to perform epidemiological analyzes using for example mortality data (or      other health indicators) to identify location-specific thresholds to be      implemented operationally.

Author Response

Round  2

Reviewer 2 Report

The methodology and the conclusions appear to be sound, but the applicability outside Switzerland is limited.  As the authors point out in the new Chapter 2, the categories “danger” and “extreme danger” do not apply in Switzerland.  Australia for example is not so fortunate, and in assessing the impact of the severe heat waves that occur in this country the Bureau of Meteorology takes into account the daily minimum temperature (which the Swiss authority does not), while humidity – which is low in most parts of the country – is not factored into heat warning criteria. 

The paper as revised has taken account of the particular points I raised in my initial review, but I remain concerned about the limited utility of the findings, which are not commensurate with the length and complexity.